# Assessment of Hematologic and Biochemical Parameters for Healthy Commercial Pigs in China

**DOI:** 10.3390/ani12182464

**Published:** 2022-09-18

**Authors:** Shuo Zhang, Bo Yu, Qing Liu, Yongjin Zhang, Mengjin Zhu, Liangyu Shi, Hongbo Chen

**Affiliations:** 1Laboratory of Genetic Breeding, Reproduction and Precision Livestock Farming, School of Animal Science and Nutritional Engineering, Wuhan Polytechnic University, Wuhan 430023, China; 2Hubei Provincial Center of Technology Innovation for Domestic Animal Breeding, Wuhan Polytechnic University, Wuhan 430023, China; 3Shenzhen Branch, Guangdong Laboratory for Lingnan Modern Agriculture & Genome Analysis Laboratory of the Ministry of Agriculture, Agricultural Genomics Institute at Shenzhen, Chinese Academy of Agricultural Sciences, Shenzhen 518000, China; 4Key Laboratory of Agricultural Animal Genetics, Breeding and Reproduction of Ministry of Education & Key Lab of Swine Genetics and Breeding of Ministry of Agriculture and Rural Affairs, Huazhong Agricultural University, Wuhan 430070, China

**Keywords:** reference interval, hematology, biochemistry, correlation analysis, commercial pigs

## Abstract

**Simple Summary:**

It is well known that hematologic and biochemical reference intervals (RIs) play a major role in defining the health state of an animal. China is home to over 50% of the world pig population, but the assessment of hematologic and biochemical parameters for healthy commercial pigs has not yet been well studied in China. Adopting hematologic and biochemical RIs from other regions may lead to misjudgments by clinicians or researchers. Therefore, the aim of this study was to evaluate the correlations between hematologic and biochemical parameters and to provide a basic reference for the establishment of blood RIs for commercial pigs in China. Since most hematologic and biochemical parameters were significantly different between nursery pigs and sows, we preliminarily established hematologic and biochemical RIs for nursery pigs and sows, respectively. Our results are useful for better assessing the health conditions of commercial pigs in China.

**Abstract:**

Hematologic and biochemical data are useful for indicating disease diagnosis and growth performance in swine. However, the assessment of these parameters in healthy commercial pigs is rare in China. Thus, blood samples were collected from 107 nursery pigs and 87 sows and were analyzed for 25 hematologic and 14 biochemical variables. After the rejection of the outliers and the detection of the data distribution, the correlations between the blood parameters were analyzed and the hematologic/biochemical RIs were preliminarily established using the 95% percentile RI. Correlation analysis showed that albumin was the hub parameter among the blood parameters investigated, and genes overlapping with key correlated variables were discovered. Most of the hematologic and biochemical parameters were significantly different between nursery pigs and sows. The 95% RIs of white blood cells and red blood cells were 7.18–24.52 × 10^9^/L and 5.62–7.84 × 10^12^/L, respectively, for nursery pigs, but 9.34–23.84 × 10^9^/L and 4.98–8.29 × 10^12^/L for sows. The 95% RIs of total protein and albumin were 43.16–61.23 g/dL and 19.35–37.86 g/dL, respectively, for nursery pigs, but 64.96–88.68 g/dL and 31.91–43.28 g/dL for sows. In conclusion, our study highlights the variability in blood parameters between nursery pigs and sows and provides fundamental data for the health monitoring of commercial pigs in China.

## 1. Introduction

Hematologic and biochemical parameters are of great importance for clinicians and researchers when assessing the health status of both humans and animals. A reference interval (RI) is a range of values based on the results of a specific percentage (usually 95%) of the healthy population [1]. Accurate and updated RIs for hematological and biochemical parameters are widely regarded as valuable standards for indicating disease diagnosis, growth performance, or nutritional condition [2,3,4]. For instance, white blood cell (WBC) counting is closely related to coronary heart disease (CHD) incidence, metabolic syndrome, and cancer mortality [5,6]. It has been reported that several blood parameters can be used as indicators for the nutritional status of pre-laying hens [4]. Among the different blood biochemical parameters, albumin is one of the most abundant blood plasma proteins and regulates the plasma oncotic pressure of blood [7]. The level of albumin is generally used as an indicator for the condition of the liver or kidneys in humans [8,9]. However, correlation analysis between albumin and other blood parameters and the bioinformatic analysis of albumin are much less common, especially in pigs.

Because blood sample collection is fairly easy to perform and repeat for the same individual during a response to a stimulus, blood sample analysis is especially useful in controlling for baseline variation in the study of porcine immune responses. Additionally, hematological characteristics are crucial traits that are associated with immune and metabolic status and diseases in pigs. Previous studies have reported that many blood parameters are heritable in swine [10,11], and serval genes have been identified by GWAS, RNA-seq, and eGWAS in pigs [12,13,14]. China has long been regarded as the largest live pig and pork consumption country in the world, and the per capita pork consumption is about 20 kg. According to the data from the United States Department of Agriculture (USDA), China produces more than 650 million pigs per year, which represents more than 50% of the global pig production. The most popular commercial pig breed in China, as well as in the world, is Duroc × (Landrace × Yorkshire) (DLY) crossbred pigs [15].

However, due to the high cost and extensive labor, there is a lack of information regarding hematologic and biochemical RIs for commercial pigs in China. So far, most information about hematologic and biochemical RIs for pigs pertain to European and North American swine breeds [16,17,18]. The Clinical and Laboratory Standards Institute (CLSI) suggests that hematologic and biochemical RIs should be established for each geographical area and specific region’s population, since RIs may vary according to location, environment, breed, age, sex, nutrition, and test method [16,19]. It is possible that adopting hematologic and biochemical RIs from non-local data may lead to the misidentification of underlying diseases and the mismanagement of health conditions. Thus, due to the great scale of the industry, it is necessary to establish hematologic and biochemical RIs for commercial pigs in China.

In the present study, we collected and examined blood samples from 194 commercial pigs in Hubei, China. Following with the bioinformatic analysis and difference determination, we established hematologic and biochemical RIs for both nursery pigs and adult sows.

## 2. Materials and Methods

### 2.1. Animals

A total of 133 crossbred nursery pigs (Duroc × Landrace × Yorkshire) and 121 adult sows (Landrace) were stochastically selected from a pig farm in Hubei province. The nursery pigs were treated with iron dextran (200 mg) injections at 7 days old and vaccinated according to the regulations of the pig farm, and blood samples were stochastically selected from 23 nursery pigs to evaluate humoral immune responses to regular vaccinations (swine fever virus, pseudorabies virus, porcine respiratory and reproductive syndrome virus, circovirus type 2, foot-and-mouth disease virus, parvovirus, Japanese encephalitis virus, streptococcus type 2, and porcine epidemic diarrhea virus). The detection was performed at Wuhan Animal Disease Diagnostic Center (Hubei, China). Pigs were closely monitored regarding their diet, water intake, body temperature, behavior, and physical characteristics. All subjects were apparently healthy in this study.

### 2.2. Blood Sample Collection and Acquisition of Blood Parameters

Blood samples were collected via the anterior vena cava using a winged infusion set (22G) connected to an EDTA-K2 blood collection tube (Shandong Ao Saite Medical Equipment, Shandong, China). All blood samples were transported to the laboratory at a temperature of 2–8 °C within 2 h after sample collection.

In total, 25 hematological parameters, including complete blood count (CBC) and white blood cell differential (WBC DIFF), were analyzed for each blood sample using an ADVIA^®^ 2120i Hematology System (Siemens Healthcare Diagnostics Inc., Erlangen, Germany) according to the manufacturer’s instructions. The measurements included white blood cells (WBCs), red blood cells (RBCs), hemoglobin (HGB), hematocrit (HCT), mean corpuscular volume (MCV), mean corpuscular hemoglobin (MCH), mean corpuscular hemoglobin concentration (MCHC), cellular hemoglobin concentration mean (CHCM), corpuscular hemoglobin (CH), red cell distribution width (RDW), hemoglobin distribution width (HDW), platelets (PLT), mean platelet volume (MPV), absolute and percentage value of neutrophil (#NEUT, %NEUT), absolute and percentage value of lymphocyte (#LYMPH, %LYMPH), absolute and percentage value of monocytes (#MONO, %MONO), absolute and percentage value of eosinophil (#EOS, %EOS), absolute and percentage value of basophilic granulocyte (#BASO, % BASO), and absolute and percentage value of unstained large cells (#LUC, %LUC). Moreover, 2 mL of blood from each sample was centrifuged at 3000× *g* for 10 min at 4 °C to obtain plasma for biochemical studies. A total of 14 biochemical parameters, including total protein (TP), albumin (ALB), aspartate transaminase (AST), alanine aminotransferase (ALT), alkaline phosphatase (ALP), total cholesterol (TC), triglyceride (TG), glucose (GLU), creatinine (CREA), high-density lipoprotein (HDL), low-density lipoprotein (LDL), blood urea nitrogen (BUN), gamma-glutamyl transpeptidase (GGT), and creatine kinase (CK) were analyzed using an automated HITEC 7100 (Hitachi, Ltd., Tokyo, Japan) according to the manufacturer’s instructions.

### 2.3. Statistics and Bioinformatics

To avoid potential adverse effects on data accuracy, both coagulated and hemolyzed blood samples were excluded. Thus, samples from 107 nursery pigs (44 males and 63 females) and 87 sows were finally used for examination. All calculations and statistical analyses were carried out in R Studio 4.1.2 (R Studio Software Inc., Boston, MA, USA), unless otherwise stated. The Shapiro–Wilk test was used to assess the normal distribution of variables. Data with nonnormal distribution were transformed using the Box–Cox power function method. Outliers for the hematologic and biochemical parameters were identified as more than 3 standard deviations (SD) from the mean and were removed.

After outlier removal, the descriptive statistics of the hematologic and biochemical parameters were analyzed using the ‘psych’ package. The descriptive statistics included unit, sample size, mean, standard deviation, median, trimmed mean, min, max, and standard error. The correlation analysis of hematologic and biochemical parameters was performed using the ‘psych’ package. A correlation heatmap of hematologic and biochemical parameters was generated using the ‘corrplot’ package.

Interaction networks of blood parameter correlations were generated using Cytoscape 3.9.0 software (St San Diego, CA, USA). A *p*-value < 0.05 was considered as the threshold value. Percentage and absolute values of the same blood parameter were not included for interaction analysis. After interaction analysis, quantitative trait loci (QTL) of ALB, HGB, HCT, #LYMPH, and %LYMPH were accessed from the Pig Quantitative Trait Locus Database (PigQTLdb) [20]. The actual genes correlated with these biochemical parameters were identified on the Ensembl website (http://asia.ensembl.org/index.html, accessed on 15 April 2022) using the ‘biomart’ tool. QTL and gene stacking maps were generated using GraphPad Prism 8.3.0 (https://www.graphpad.com/scientific-software/prism/, accessed on 20 April 2022) and a Venn diagram of the candidate genes in QTL regions was generated using RStudio 4.1.2 (RStudio PBC, Boston, USA). The gene set enrichments of candidate genes were analyzed on the Kyoto Encyclopedia of Genes and Genomes (KEGG) website (https://www.genome.jp/kegg/, accessed on 20 April 2022).

Differences between nursery pigs and sows were analyzed by two-tailed unpaired Student’s *t*-tests, and differences between nursery pigs at different ages were examined by one-way ANOVA, followed by a post hoc Duncan test.

Hematologic and biochemical RIs were established using Analyze-It 5.30 (Analyze-It Software, Leeds, UK) according to guidelines of CLSI. Reference intervals were determined using the 95% reference intervals (2.5th and 97.5th percentile) for each hematological and biochemical parameter.

## 3. Results

Blood parameter data were analyzed after quality control. After outlier removal (more than 3 × SD), blood parameter data were used for the subsequent analysis. Details of the descriptive statistics of the 25 hematological parameters and 14 plasma biochemical parameters are presented in Appendix A.

### 3.1. Correlations of Blood Parameters and Bioinformatic Analysis

A total of 399 pairs of blood parameters (hematology and plasma biochemistry) showed significant correlation (*p* < 0.05). Among these, 41 pairs of blood parameters had a strong correlation (|r| > 0.5, *p* < 0.05), with 31 pairs of blood parameters being positively correlated and 10 pairs negatively correlated (Figure 1A). TC had the strongest positive correlation with LDL (r = 0.95), followed by the correlation between MCV and CH (r = 0.93). Meanwhile, %NEUT and %LYMPH had the strongest negative correlation (r = −0.86), followed by the correlation between BUN and GLU (r = −0.69).

Based on the correlational interaction results, three clusters of blood parameters, including hematologic analytes CBC, hematologic analytes WBC DIFF, and biochemical analytes, were identified and the correlation networks were constructed (Figure 1B). As shown in the figure, there were moderate correlations between the parameters of CBCs. In detail, MCV was negatively correlated with RDW (r = −0.57), while MCHC was positively correlated with MPV (r = 0.4). As for the cluster of WBC DIFF, #NEUT were negatively correlated with #LYMPH and #BASO but positively correlated with #MONO. #NEUT and %LYMPH were strongly negatively correlated (r = −0.86). There was a moderately positive correlation between #LYMPH and #BASO (r = 0.45). For the plasma biochemical traits, most of the parameters were negatively correlated, with the strongest negative correlation between TP and GLU (r = −0.72), followed by the correlation between GLU and BUN (r = −0.69). More importantly, ALB had a correlation with most of the parameters in the CBC cluster. Among them, ALB was positively correlated with HCT and HGB (r = 0.54 and 0.49, respectively).

Bioinformatic analyses of ALB and its strongly correlated parameters (HGB, HCT, #LYMPH and %LYMPH) were further performed. The QTL results and actual genes correlated with theses parameters are presented in Figure 1C. There were 285 HCT-related QTLs and 232 correlated genes, distributed in all 19 chromosomes and located more frequently on chromosomes 7, 9, and 14. Based on the QTL results, the number of overlapping genes related to HGB and HCT was 56 (Figure 1D). Furthermore, the number of overlapping genes related to #LYMPH, HGB, and HCT was 10. However, there was no overlapping gene between ALB and the other four blood parameters. The number of overlapping genes related to #LYMPH and HGB, #LYMPH and HCT, and #LYMPH and %LYMPH were 10, 26, and 2, respectively. Details of QTLs, related genes, and KEGG results are shown in Appendix A.

### 3.2. Determination of Difference in Blood Parameters between Nursery Pigs and Sows

To assess the influence of age on blood parameters in nursery pigs, we analyzed the significant differences in blood parameters between nursery pigs at different ages. There were no significant differences (*p* > 0.05) in six hematologic parameters among nursery pigs at different ages (Appendix A). Among the other 19 hematologic parameters, only RDW showed significant differences (*p* < 0.05) between all nursery pigs at different ages. Furthermore, although age had a significant effect in nursery pigs, only TP and ALB were significantly different between all ages (Appendix A).

We then determined the significant differences in blood parameters between nursery pigs and sows. As expected, 19 out of the 25 hematologic parameters showed significant differences (*p* < 0.05) between nursery pigs and sows (Table 1). Moreover, 12 out of the 14 biochemical parameters were significantly different between nursery pigs and sows (Table 2). Thus, it is necessary to establish RIs of blood parameters for nursery pigs and sows separately.

### 3.3. Assessment of Blood Parameter RIs in Nursery Pigs and Sows

Table 3 displays the 95% RIs and 90% CIs for the lower and upper reference limits of the hematologic parameters for nursery pigs. The 95% reference intervals of WBC, RBC, HGB, and HCT for nursery pigs were 7.18–24.52 × 10^9^/L, 5.62–7.84 × 10^12^/L, 92.20–135.20 g/dL, and 31.13–45.49%, respectively. The RIs and other statistical values of the biochemical parameters for nursery pigs are presented in Table 4. The 95% RIs of TP, ABL, AST, and ALT for nursery pigs were 43.16–61.23 g/dL, 19.35–37.86 g/dL, 27.20–89.90 U/L, and 26.00–72.10 U/L, respectively.

The RIs and other statistical values of the hematologic parameters for sows are displayed in Table 5. The 95% RIs of WBC, RBC, HGB, and HCT were 9.34–23.84 × 10^9^/L, 4.98–8.29 × 10^12^/L, 92.80–140.30 g/dL, and 32.11–47.85%, respectively, for sows. The RIs and other statistical values of the biochemical parameters for sows are shown in Table 6. The 95% RIs of TP, ABL, AST, and ALT for sows were 64.96–88.68 g/dL, 31.91–43.28 g/dL, 20.40–107.70 U/L, and 26.80–60.10 U/L, respectively.

## 4. Discussion

The establishment of hematologic and biochemical RIs is important for researchers and veterinarians so that they can better understand the health status of individuals. According to the CLSI regulations, each laboratory should develop its own hematologic and biochemical RIs for a specific population or herd. However, due to the costly and time-consuming nature of this process, there is a dearth of information on the hematologic and biochemical RIs of commercial pigs in China, which account for nearly half of the world pig population.

We first evaluated all the data we collected before the assessment of the hematologic and biochemical RIs for commercial pigs in China. In total, 25 hematologic parameters and 14 biochemical parameters were analyzed after outlier removal. Among the blood parameters, the correlation between TC and LDL was the highest (*r* = 0.95), which was understandable. TC includes both LDL and HDL, and LDL is the main source of cholesterol buildup [21]. TC and HDL were consistently strongly correlated (*r* = 0.83) in this study. In regards to negative correlation, neutrophils and the percentage of lymphocytes were strongly correlated (*r* = −0.86). Neutrophils and lymphocytes make up the largest portion of white blood cells, so it is obvious why a higher number of neutrophils would decrease the percentage of lymphocytes in pigs. Interestingly, there were only two ALB-related QTLs located on chromosome 6 and only one correlated gene (*GRHL3*), while there were 285 HCT-related QTLs and 232 correlated genes. *GRHL3* encodes a number of transcription factors that are involved in neural tube closure and wound repair [22]. However, the relationship between ALB and GRHL3 proteins has not yet been established.

Nursery pigs undergo rapid changes in their bodies and immune systems, which may have an impact on the variation in their hematologic and biochemical parameters. Therefore, we compared the blood parameters in pigs from 20 to 50 days old. Unexpectedly, only 3 out of 39 blood parameters showed significant differences in all the nursery pigs between different ages. However, according to the Duncan’s multiple range tests, 23 out of 39 blood parameters were significantly different between 10-day-old nursery pigs and 40-day-old nursery pigs, indicating that these blood parameters changed significantly with a large enough age difference. Similar results regarding differences in the blood parameters of nursery pigs have been found in other publications [23,24], indicating that age is an essential factor for hematologic and biochemical parameters in nursery pigs [18]. It has been reported that new-born piglets and adult pigs are remarkably different [25]. Thus, we also checked the difference in blood parameters between nursery pigs and sows. As expected, 31 out of 39 blood parameters were significantly different between nursery pigs and sows. Indeed, several studies have demonstrated that most hematologic and biochemical parameters are significantly different between nursery pigs and adult pigs [18,26,27].

In the present study, the value of WBC was 7.18–24.52 × 10^9^/L for nursery pigs and 9.34–23.84 × 10^9^/L for sows. However, the reference interval of WBC is 6.0–21.7 × 10^9^/L for Ontario piglets, 13.7–17.3 × 10^9^/L for 5-week-old pigs in Wisconsin, and 5.6–18.5 × 10^9^/L for 30-day-old piglets in Italy [18,25,28]. The range of WBC values was much wider in our study compared to other studies. This was possibly due to the fact that the immune system undergoes rapid changes in nursery pigs, as shown in this study, and pigs aged 20 to 50 days were subjected to analysis. In contrast, the WBC values for sows in this study were similar to those of other studies. For instance, the reference interval of WBC is 10.12–22.24 × 10^9^/L for Danish sows at mid-gestation [29]. The value of RBC was 5.62–7.84 × 10^9^/L for nursery pigs in our study. The reference interval of RBC is 4.8–7.3 × 10^12^/L for Ontario piglets and 4.08–8.17 × 10^12^/L for 30-day-old piglets in Italy [26]. The value of RBC was 4.98–8.29 × 10^12^/L for sows in our study, while the reference interval of RBC is 4.98–7.50 × 10^12^/L for Danish sows at mid-gestation [25,29].

ALB plays an important role in maintaining osmotic pressure and transporting numerous substances in the blood [30]. The value of ALB was 19.35–37.86 g/dL for nursery pigs in our study, while the reference interval of ALB is 24.9–46.0 g/dL for Ontario piglets, which is higher than our result [18]. However, the reference interval of ALB is only 1.9–4.0 g/dL, which is much lower than our result and the results reported by Amanda et al. [18,25]. Moreover, the reference intervals of AST also differ between studies: the value was 27.20–89.90 U/L in our study, while it is 18.0–83.5 U/L for Ontario piglets and 13.00–65.00 U/L for 30-day-old piglets in Italy [18,25]. The study of biochemical parameter RIs for sows is limited. The study reported by A.R.W. Elbers et al. provided an RI of ABL in sows of 24.1–39.3 g/L [31], which is substantially different from the RI of ALB in sows (31.91–43.28 g/dL) calculated in the present study. It should be emphasized that the variation in CK was the widest in both the present study and previous reports. The variation between maximum and minimum was nine-fold in nursery pigs, while it expanded to 31-fold in sows (Table 4 and Table 6). As far as we know, variations in CK (reference values) are also the widest in other studies, in spite of the different measurement instruments used. The value ranges are 1.75–99.99 µkat/L for healthy Yucatan micropigs at the age of 20 ± 4 weeks [17] and 111.0–4918.0 U/L for Ontario commercial nursing pigs [18]. Additionally, the CK variation is widest in both the reference values (359–28,155 U/L) and 95% RI (0–10101 U/L) for Norwegian crossbreed grower pigs at the age of 12–16 weeks [16].

Taken together, several blood parameters are subject to broad variations, indicating a dramatic physiological evolution in healthy pigs. According to the comparison analysis between our data and previously published data, region, age, and breed are essential factors that should be considered when assessing blood parameter RIs. China is a large country with highly diverse geographical and environmental features; thus, a limitation of this study was that we only included nursing pigs and sows from one region in China for RI assessment. Therefore, to establish more accurate blood parameter RIs for commercial pigs in China, studies with larger sample sizes from more commercial farms are necessary.

## 5. Conclusions

To conclude, this preliminary study showed that ALB was a hub parameter among the blood variables, due to its correlation with most hematologic and biochemical parameters. We pointed out the significant differences in the hematologic and biochemical variables between nursery pigs and sows. Thus, the hematologic and biochemical RIs for nursery pigs and sows were assessed individually. Because China is home to over 50% of the world pig population, our data provide a basic reference for veterinarians and breeding researchers when assessing the health status of commercial pigs in China.

## Figures and Tables

**Figure 1 animals-12-02464-f001:**
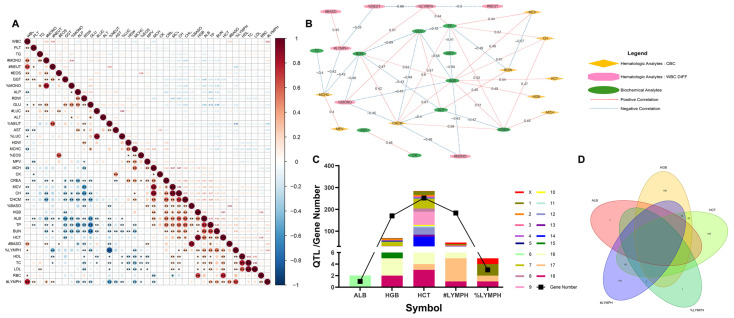
Correlation and bioinformatic analysis of blood parameters. (**A**) Heatmap showing correlation between different blood parameters. Pearson’s correlation coefficients are shown on the top-right triangle, *p*-values (* *p* < 0.05, ** *p* < 0.01) are shown on the bottom left triangle. (**B**) Network analysis of correlation between different blood parameters. Percentage and absolute values of the same blood parameter were not included in the interaction analysis. *p*-value < 0.05 and |r| ≥ 0.4 were considered as the threshold values. (**C**) Quantitative trait locus and gene locations of ALB, HGB, HCT, #LYMPH, and %LYMPH. (**D**) Venn diagram of candidate genes in QTL regions of ALB, HGB, HCT, #LYMPH, and %LYMPH. WBC = white blood cell, RBC = red blood cell, HGB = hemoglobin, HCT = hematocrit, MCV = mean corpuscular volume, MCH = mean corpuscular hemoglobin, MCHC = mean corpuscular hemoglobin concentration, CHCM = cellular hemoglobin concentration mean, CH = corpuscular hemoglobin, RDW = red cell distribution width, HDW = hemoglobin distribution width, PLT = platelets, MPV = mean platelet volume, NEUT = neutrophil, LYMPH = lymphocyte, MONO = monocytes, EOS = eosinophil, BASO = basophilic granulocyte, LUC = unstained large cells, TP = total protein, ALB = albumin, AST = aspartate transaminase, ALT = alanine aminotransferase, ALP = alkaline phosphatase, TC = total cholesterol, TG = triglyceride, GLU = glucose, CREA = creatinine, HDL = high-density lipoprotein, LDL = low-density lipoprotein, BUN = blood urea nitrogen, GGT = gamma-glutamyl transpeptidase, CK = creatine kinase, ‘#’ = absolute value, ‘%’ = percentage value.

**Table 1 animals-12-02464-t001:** Comparison of hematologic parameters between nursery pigs and sows.

Variable	Unit	Nursery Pigs	Sows	Variable	Unit	Nursery Pigs	Sows
**WBC**	10^9^/L	15.85 ± 4.35	16.59 ± 3.62	**%NEUT**	%	38.14 ± 8.87	31.98 ± 9.35 **
**RBC**	10^12^/L	6.75 ± 0.56	6.90 ± 0.82	**%LYMPH**	%	45.34 ± 9.84	55.70 ± 8.96 **
**HGB**	g/dL	110.30 ± 10.72	116.60 ± 11.90 **	**%MONO**	%	8.16 ± 3.34	5.518 ± 2.13 **
**HCT**	%	37.12 ± 3.56	39.98 ± 3.94 **	**%EOS**	%	4.52 ± 2.06	4.54 ± 2.47
**MCV**	fL	55.00 ± 3.88	58.46 ± 3.53 **	**%BASO**	%	0.37 ± 0.11	0.50 ± 0.29 **
**MCH**	pg	16.36 ± 1.23	17.03 ± 1.59 **	**%LUC**	%	2.17 ± 1.03	1.59 ± 1.05 **
**MCHC**	g/dL	297.10 ± 8.37	291.10 ± 15.15 **	**#NEUT**	10^9^/L	5.96 ± 1.95	5.31 ± 1.92 *
**CHCM**	g/Dl	323.70 ± 9.95	342.30 ± 8.99 **	**#LYMPH**	10^9^/L	7.12 ± 2.66	9.27 ± 2.50 **
**CH**	pg	17.78 ± 1.48	19.92 ± 1.37 **	**#MONO**	10^9^/L	1.30 ± 0.65	0.93 ± 0.44 **
**RDW**	%	19.99 ± 2.82	17.783 ± 0.90 **	**#EOS**	10^9^/L	0.79 ± 0.55	0.74 ± 0.39
**HDW**	g/dL	21.07 ± 1.69	20.93 ± 2.28	**#BASO**	10^9^/L	0.06 ± 0.02	0.08 ± 0.05 **
**PLT**	10^9^/L	287.60 ± 155.90	227.50 ± 106.20 **	**#LUC**	10^9^/L	0.34 ± 0.17	0.25 ± 0.16 **
**MPV**	fL	9.54 ± 1.33	9.64 ± 1.54				

Data are represented as mean ± standard deviation, * *p* < 0.05, ** *p* < 0.01.

**Table 2 animals-12-02464-t002:** Comparison of biochemical parameters between nursery pigs and sows.

Variable	Unit	Nursery Pigs	Sows	Variable	Unit	Nursery Pigs	Sows
**TP**	g/dL	51.20 ± 4.54	75.60 ± 5.98 **	**GLU**	mg/dL	90.63 ± 15.63	51.13 ± 15.63 **
**ALB**	g/dL	28.60 ± 4.66	37.59 ± 2.86 **	**CREA**	mg/dL	81.26 ± 17.72	115.70 ± 35.67 **
**AST**	U/L	47.99 ± 5.85	42.88 ± 22.73 **	**HDL**	mg/dL	43.433 ± 11.10	45.30 ± 7.69
**ALT**	U/L	44.11 ± 1.52	40.08 ± 8.40 **	**LDL**	mg/dL	56.75 ± 24.56	64.05 ± 11.10 **
**ALP**	U/L	48.35 ± 25.85	16.72 ± 16.20 **	**BUN**	mg/dL	3.26 ± 1.00	7.63 ± 2.51 **
**TC**	mg/dL	80.21 ± 24.15	88.57 ± 13.25 **	**GGT**	U/L	93.50 ± 50.43	59.50 ± 38.36 **
**TG**	mg/dL	45.01 ± 15.69	42.12 ± 20.15	**CK**	U/L	735.10 ± 509.10	1399.40 ±1723.90 **

Data are represented as mean ± standard deviation, ** *p* < 0.01.

**Table 3 animals-12-02464-t003:** Hematology values and RIs for nursery pigs.

Variable	Unit	N	95% RI	90% CI Lower Limit	90% CI Upper Limit
**WBC**	10^9^/L	106	7.18–24.52	5.99–8.37	23.34–25.71
**RBC**	10^12^/L	106	5.62–7.84	5.46–5.78	7.69–7.98
**HGB**	g/dL	107	92.20–135.20	90.30–94.10	130.90–139.90
**HCT**	%	107	31.13–45.49	30.52–31.78	44.01–47.09
**MCV**	fL	106	47.28–62.74	46.22–48.33	61.68–63.80
**MCH**	pg	107	13.90–18.81	13.57–14.24	18.48–19.15
**MCHC**	g/dL	107	280.50–313.80	278.20–282.70	311.50–316.10
**CHCM**	g/dL	107	303.80–343.50	301.10–306.50	340.80–346.20
**CH**	pg	107	14.87–20.68	14.47–15.27	20.28–21.08
**RDW**	%	107	16.34–27.88	16.05–16.65	25.74–30.97
**HDW**	g/dL	106	18.29–25.00	18.00–18.58	24.29–25.77
**PLT**	10^9^/L	106	38.74–656.85	22.03–59.34	594.88–721.45
**MPV**	fL	106	6.90–12.180	6.53–7.26	11.81–12.54
**%NEUT**	%	105	20.46–55.81	18.02–22.89	53.38–58.25
**%LYMPH**	%	105	22.73–60.87	16.32–27.51	59.13–62.55
**%MONO**	%	104	1.51–14.82	0.59–2.43	13.90–15.74
**%EOS**	%	106	1.77–9.99	1.59–1.98	8.80–11.35
**%BASO**	%	107	0.18–0.62	0.16–0.20	0.58–0.66
**%LUC**	%	107	0.60–4.68	0.49–0.73	4.23–5.16
**#NEUT**	10^9^/L	105	2.07–9.85	1.54–2.61	9.32–10.39
**#LYMPH**	10^9^/L	107	1.84–12.42	1.12–2.56	11.69–13.14
**#MONO**	10^9^/L	106	0.25–2.81	0.17–0.35	2.56–3.08
**#EOS**	10^9^/L	105	0.22–2.31	0.19–0.25	1.91–2.8
**#BASO**	10^9^/L	105	0.02–0.12	0.02–0.03	0.11–0.13
**#LUC**	10^9^/L	106	0.09–0.77	0.07–0.11	0.69–0.85

CI = confidence interval.

**Table 4 animals-12-02464-t004:** Biochemical values and RIs for nursery pigs.

Variable	Unit	N	95% RI	90% CI Lower Limit	90% CI Upper Limit
**TP**	g/dL	106	43.16–61.23	42.35–44.00	59.77–62.75
**ALB**	g/dL	107	19.35–37.86	18.22–20.49	36.72–38.99
**AST**	U/L	106	27.20–89.90	25.90–28.70	81.10–100.40
**ALT**	U/L	106	26.00–72.10	24.60–27.60	67.40–77.20
**ALP**	U/L	103	16.90–120.20	15.20– 18.80	104.40–138.70
**TC**	mg/dL	107	50.24–143.46	48.27–52.38	129.20–161.12
**TG**	mg/dL	106	20.03–82.24	18.05–22.17	76.29–88.53
**GLU**	mg/dL	106	59.59–121.67	55.77–63.42	117.85–125.50
**CREA**	mg/dL	107	46.10–116.40	41.80–50.40	112.10–120.80
**HDL**	mg/dL	107	29.74–74.45	28.84–30.70	66.81–84.63
**LDL**	mg/dL	107	27.32–120.98	25.56–29.26	106.84–137.94
**BUN**	mg/dL	106	1.61–5.57	1.47–1.76	5.22–5.94
**GGT**	U/L	107	29.60–225.30	26.00–33.60	119.30–254.70
**CK**	U/L	104	247.10–2229.00	225.90–271.10	1805.80–2805.10

CI = confidence interval.

**Table 5 animals-12-02464-t005:** Hematology values and RIs for sows.

Variable	Unit	N	95% RI	90% CI Lower Limit	90% CI Upper Limit
**WBC**	10^9^/L	87	9.34–23.84	8.26–10.42	22.76–24.92
**RBC**	10^12^/L	86	4.98–8.29	4.53–5.37	8.12–8.46
**HGB**	g/dL	86	92.80–140.30	89.20–96.30	136.80–143.90
**HCT**	%	86	32.11–47.85	30.93–33.29	46.67–49.03
**MCV**	fL	86	52.75–66.51	52.10–53.43	64.92–68.27
**MCH**	pg	87	14.78–20.67	14.55–15.02	19.84–21.66
**MCHC**	g/dL	87	265.8–0325.80	262.80–268.90	318.70–332.40
**CHCM**	g/dL	87	325.50–361.40	323.30–327.80	358.20–364.60
**CH**	pg	87	17.80–23.01	17.57–18.05	22.38–23.72
**RDW**	%	87	15.98–19.59	15.71–16.25	19.32–19.86
**HDW**	g/dL	86	17.81–26.43	17.51–18.13	25.09–28.13
**PLT**	10^9^/L	87	38.38–580.95	26.37–52.22	541.13–621.94
**MPV**	fL	86	7.23–13.25	6.98–7.50	12.52–14.06
**%NEUT**	%	87	13.29–50.67	10.51–16.08	47.89–53.46
**%LYMPH**	%	87	37.79–73.61	35.12–40.46	70.94–76.28
**%MONO**	%	86	2.22–10.71	1.92–2.55	9.68–11.83
**%EOS**	%	86	1.50–11.05	1.30–1.73	9.45–12.93
**%BASO**	%	86	0.18–1.43	0.16–0.20	1.14–1.83
**%LUC**	%	87	0.31–4.84	0.25–0.38	3.99–5.85
**#NEUT**	10^9^/L	87	1.48–9.14	0.90–2.05	8.57–9.71
**#LYMPH**	10^9^/L	87	4.26–14.27	3.51–5.00	13.52–15.01
**#MONO**	10^9^/L	87	0.35–2.10	0.31–0.40	1.82–2.421
**#EOS**	10^9^/L	86	0.22–1.74	0.18–0.26	1.51–1.99
**#BASO**	10^9^/L	86	0.02–0.23	0.02–0.027	0.19–0.27
**#LUC**	10^9^/L	87	0.05–0.72	0.04–0.06	0.61–0.86

CI = confidence interval.

**Table 6 animals-12-02464-t006:** Biochemical values and RIs for sows.

Variable	Unit	N	95% RI	90% CI LowerLimit	90% CI UpperLimit
**TP**	g/dL	87	64.96–88.68	63.83–66.12	86.74–90.70
**ALB**	g/dL	86	31.91–43.28	31.18–32.63	42.55–44.00
**AST**	U/L	87	20.40–107.70	19.20–21.70	88.90–134.50
**ALT**	U/L	87	26.80–60.10	25.70–28.10	56.60–63.80
**ALP**	U/L	87	1.20–65.90	0.80–1.60	53.80–80.20
**TC**	mg/dL	87	58.84–112.04	53.65–63.59	109.44–114.57
**TG**	mg/dL	86	12.78–92.60	10.84–14.98	83.58–102.33
**GLU**	mg/dL	87	20.06–82.19	16.11–24.00	78.25–86.14
**CREA**	mg/dL	87	75.10–228.70	72.60–77.80	194.30–284.40
**HDL**	mg/dL	87	30.01–60.59	28.07–31.96	58.65–62.53
**LDL**	mg/dL	87	42.00–86.11	39.19–44.8	83.30–88.91
**BUN**	mg/dL	87	3.79–13.89	3.50–4.12	12.78–15.10
**GGT**	U/L	87	20.80–157.30	18.80–23.00	133.20–187.10
**CK**	U/L	86	268.00–8353.10	240.60–300.20	5097.10–15910.20

CI = confidence interval.

## Data Availability

The data that support the findings of this study are available from the corresponding author upon reasonable request.

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
