# Peer review of "Assessment of Hematologic and Biochemical Parameters for Healthy Commercial Pigs in China"

_animals, 2022, doi:10.3390/ani12182464_

Round 1

Reviewer 1 Report

The manuscript is interesting and novel. It is recommended to modify the following points:

Line 32-49: The abstract exceeds the maximum of 200 words indicated in the author's guide (Microsoft word template). Modifications needed

Line 50: Like the title, keywords are used to increase the search level of the manuscript in databases, so it is recommended that most of these are words that do not appear in the title. Could change some

Line 58: change [2-4] by [24]

Line 73: change [12-14] by [12–14]

Line 74-73: indicate the per capita consumption of pork in this country

Line 81: change [16-18] by [16–18]

Line 109: 2-8 °C

Line 109: 2 h

Line 124: …Moreover, 2 mL….

Line 124: 3,000xg

Line 158: agas ? ages?

Line 171: the figure was not previously described in the text

Line 171: the figures are used as support to visualize the results obtained in this research. Although in figure 1 different colors are used to make schematic differences of the results, the font size is so small that it is difficult to observe what was obtained. Modifications Needed (Could the information be separated into two figures?)

Line 191: … correlation (p < 0.05)

Line 193,199,213,216: figures must appear after being mentioned

Line 225: (p > 0.05)

Line 227,235: (p < 0.05)

Line 238,240: parameters between nursery pigs

Line 238: remove under table... Data are represented as mean ± standard deviation, *p < 0.05, **p < 0.01.

Line 240: remove under table... Data are represented as mean ± standard deviation, **p < 0.01.

Line 244: …95% RIs and 90%...

Line 245,248,252,255: The 95% RIs of…?

Line 257,262,264,265 : the meaning of the abbreviations must be described under the table

Line 277: r = 0.95

Line 279: r = 0.83

Line 290: …from 20 to 50 days old

Line 304: 6.0-21.7

Line 305: 5.6-18.5*

Line 308: from 20 to 50 days

Line 311: delete space…22.24*

Line 313: 4.8-7.3

Line 313: 4.08-8.17*

Line 315: 7.50*

Line 319: 24.9-46.0

Line 323: 18.0-83.5

Line 326: 24.1-39.3

Line 346,348: (p < 0.05)…(p < 0.01)

Line 373,375: According to the author guide the journal name of the reference must be abbreviated. Check format in the following references

Line 374: change 83-90, doi by 83–90. Doi (Check format in the following references)

Line 378: scientific names should be written in cursive text format

Line 381: scientific names should be written in cursive text format

Line 420: complete the list of reference authors

Line 431: complete the list of reference authors

Line 450: the title appears between square brackets, is it correct?

Note: could the weather affect the hematological and biochemical parameters? Could you consider the season of the year (summer and winter) as another factor in the study?

Reviewer 2 Report

This paper describes essential data for pig health monitoring, but some data must be well represented, and answers are needed to validate these data better. 

Did the authors assessed parasite infections that could interfere with hematologic data? 

The authors described at Line 96 that pigs received iron injection before sample collection. How do they managed the influence of iron injection on haematological parameters as former studies noted that injection of iron dextran causes a significant increase in haematological parameters in animals (See Allan J, Plate P, Van Winden S. The Effect of Iron Dextran Injection on Daily Weight Gain and Haemoglobin Values in Whole Milk Fed Calves. Animals (Basel). 2020 May 14;10(5):853. DOI: 10.3390/ani10050853.) ?

L164 165: This is a description of methods and needs to be moved to this section

We have noted a contradiction between information on Line 232 and 293-294. Authors declared first that there is no need to establish RIs based on age group but noted significant changes in blood parameters according to age. How do You justify this? 

The authors described data in tables 4, 5, and 6 with an important range. For example, in table 6, CK varied from 268 to 8353. This is too important for data that will be used as a reference value.  This point needs to be discussed widely in the discussion section.

Overall, we propose to change the title as "Assessment of hematologic and Biochemical parameters for Healthy Commercial Pigs in China". The study as it has been design is not suitable for reference value determination. Also, some intervals are too wide to be used as reference data. 

Round 2

Reviewer 1 Report

Although most of the requested changes were addressed, it is necessary to review the following:

Line 304: change 5.62-7.84 by 5.62–7.84

Line 378: change 643-657 by 643–657

Line 380: according to the author guide the reference journals should be appear in abbreviate format (Fish Shellfish Immunol.)

Line 381: change 182-188 by 182–188

Line 384: according to the author guide the reference journals should be appear in abbreviate format (J. Zoo Wildl. Med.)

Line 384: change 422-429 by 422–429

Line 387: change 1932-1937 by 1932–1937

Line 389: according to the author guide the reference journals should be appear in abbreviate format (Circ. J. 2004)

Line 390: change 892-897 by 892–897

Line 389: Idem (J. Pediatr. Surg.)

Line 389: change 597-599 by 597–599

Line 393: Idem (Ann. Hepatol.)

Line 393: change 547-560 by 547–560

Line 395: Idem (J. Am. Soc. Nephrol.)

Line 404: Idem (BMC Genom.)

Line 408: Idem (BMC Genom.)

Line 410: Idem (J. Anim. Sci.)

Line 411: change 5028-5041 by 5028–5041

Line 412: change the title by "Transcriptome analysis identifies biological pathways and candidate genes for feed efficiency in DLY pigs"

Line 415: Idem (Vet. Clin. Pathol.)

Line 415: change 221-226 by 221–226

Line 417: Idem (Lab. Anim.)

Line 417: change 368-373 by 368–373

Line 419: Idem (Can. Vet. J.)

Line 420: change 371-376 by 371–376

Line 423: Idem (SAGE Open Med.)

Line 426: Idem (Nucleic Acids Res.)

Line 426: change D871-879 by D871–879

Line 430: Idem (J. Am. Coll. Cardiol.)

Line 430: change 35-41 by 35–41

Line 432: Idem (Dev. Biol.)

Line 432: change 263-272 by 263–272

Line 435: Idem (BMC Genet.)

Line 437: Idem (Acta Vet. Scand.)

Line 438: change 381-393 by 381–393

Line 441: Idem (BMC Vet. Res.)

Line 443: Idem (Can. J. Comp. Med.)

Line 443: change 390-393 by 390–393

Line 445: Idem (J. Vet. Diagn. Investig.)

Line 446: change 561-567 by 561–567

Line 449: Idem (J. Vet. Diagn. Investig.)

Line 450: change 438-443 by 438–443

Line 451: Idem (Acta Vet. Scand.)

Line 453: Idem (Postepy Hig. Med. Dosw.)

Line 454: change 17-36 by 17–36

Line 456: Idem (Vet. Q.)

Line 456: change 127-130 by 127–130
